# Effect of Dupilumab on Sexual Desire in Adult Patients with Moderate to Severe Atopic Dermatitis

**DOI:** 10.3390/medicina58121708

**Published:** 2022-11-23

**Authors:** Maddalena Napolitano, Gabriella Fabbrocini, Sara Kastl, Teresa Battista, Adriana Di Guida, Fabrizio Martora, Vincenzo Picone, Virginia Ventura, Cataldo Patruno

**Affiliations:** 1Department of Medicine and Health Sciences Vincenzo Tiberio, University of Molise, 86100 Campobasso, Italy; 2Section of Dermatology, Department of Clinical Medicine and Surgery, University of Naples Federico II, 80131 Napoli, Italy; 3Centro di Psicologia e Analisi Transazionale, 20019 Milan, Italy; 4Department of Health Sciences, University Magna Graecia of Catanzaro, 88100 Catanzaro, Italy

**Keywords:** atopic dermatitis, sexual disfunction, SDI-2

## Abstract

*Background*: Atopic dermatitis (AD) is a chronic inflammatory condition causing itching skin, with a significant psychosocial impact on patients and relatives. AD affects 15 to 30% of children and 2 to 10% of adults. AD significantly affects patients’ quality of life (QoL) given the chronicity and symptoms of the disease. Most AD patients have reported that the disease affects daily life, resulting in limited social contact and a strong impact on sexual health (SH), especially in moderate–severe cases. *Materials and methods*: We performed a prospective study from 1 May 2020 to 1 May 2022; the aim of the study was to evaluate the impact of moderate to severe AD on sexual desire, seduction, and partner relationships, and describe how it varies following dupilumab therapy. We used the Sexual Desire Inventory-2 (SDI-2), a validated instrument consisting of 14 items; moreover, we used a second questionnaire with eight items, an unvalidated instrument created specifically for this study, to obtain the assessment of the influence of AD on the body image, sexuality, and self-perception of those affected. *Results and Conclusions*: The impact of AD on sexual desire assessed using SDI-2 showed a significant improvement in both sexes during dupilumab treatment from the baseline to W4 and W16. Similar results were obtained with our questionnaire.

## 1. Introduction

Atopic dermatitis (AD) is a chronic inflammatory skin disease characterized by eczema and intense itching; skin barrier impairment and overregulation of type 2 immune responses are considered the main pathogenetic features [1]. AD affects 15 to 30% of children and 2 to 10% of adults [2]. Evidence suggests that the prevalence of both persistent and new-onset cases in adulthood is increasing [3].

Because of the chronicity and discomfort of symptoms, AD significantly affects the patients’ quality of life (QoL) [4]. Most patients with AD have reported that the disease affects their daily life, resulting in limited social contact and a strong impact on sexual health (SH) [5], especially in moderate-to-severe cases.

The World Health Organization defines SH as “a state of physical, emotional, mental, and social well-being in relation to sexuality” and further specifies that sexual health includes a “positive and respectful approach to sexuality and sexual relationships” [6]. AD factors potentially interfering with SH include physical appearance, reduced self-esteem, and unpleasant sensations in erogenous areas due to itching and pain [7]. Indeed, AD patients are more likely to experience hypersensitivity reactions or irritation in response to sweat, seminal fluid, lubricants, or latex [8]. In a cross-sectional study conducted in 13 European countries, item 9 of the Dermatology Life Quality Index (DLQI) was used to collect information on sexual impact in 3485 AD adult patients affected by chronic skin diseases such as hidradenitis suppurativa, prurigo, blistering disorder, psoriasis, and urticaria and eczema [4]. The question was “Over the last week, how much has your skin caused any sexual difficulties?”, with the possible answers “very much” (3), “a lot” (2), “a little” (1), “not at all/not relevant” (0). Around 29% of 448 AD patients experienced sexual difficulties. In addition, AD has been also shown to negatively impact partners, due the fear of causing pain during sexual activity or transmitting AD to their child, or because the AD could induce a reduction in the sexual desire of the patient [9]. Sexual desire has been broadly described as “the sum of the forces that lean us toward and push us away from sexual behaviour”, or as “a psychological state subjectively experienced by the individual as an awareness that he or she wants or wishes to attain a (presumably pleasurable) sexual goal that is currently unattainable” [10].

Dupilumab is a monoclonal antibody directed against the α subunit of the interleukin (IL) 4 receptor, and it blocks IL 4- and 13-mediated signaling [11]. It is indicated in the treatment of moderate-to-severe AD, asthma, and chronic sinusitis associated with nasal polyposis [11]. Dupilumab demonstrated a significant improvement in disease severity, pruritus, and QoL in patients with AD with a high safety profile, even in the frailest patients, both in clinical trials and real-life studies [5,11,12,13].

The aim of this study was to evaluate the impact of moderate-to-severe AD on sexual desire, seduction, and partner relationships, and describe how it varies following dupilumab therapy.

## 2. Materials and Methods

A prospective study was performed at the Dermatology Unit of the University of Naples Federico II, from 1 May 2020 to 1 May 2022. The recruited patients were affected by moderate-to-severe AD and treated with dupilumab based on Italian Medicines Agency indications. Inclusion criteria were age 18–65 years, Eczema Area Severity Index (EASI) ≥ 24, and contraindication/side effects/failure to respond to cyclosporine (CsA). Patients affected by other skin or systemic diseases with a possible impact on SH were excluded. Sociodemographic variables (age, sex, family status), clinical data (medical history and AD management, AD location and clinical phenotype), comorbidities (atopic and non-atopic), and dupilumab adverse events (AEs) were collected for each patient. Disease severity was assessed at baseline, 4 weeks (W4) and W16 after starting dupilumab therapy. EASI (range 0–72), Pruritus Numeric Rating Scale (P-NRS) (range 0–10) evaluated as peak score during the past 7 days, and DLQI score (range 0–30) were used for the assessment. To evaluate the impact of both AD and dupilumab treatment on the features relating to sexuality that we aimed to investigate, we used the Sexual Desire Inventory-2 (SDI-2), a validated instrument consisting of 14 items (9 with responses on an 8-point Likert scale), which measures the multidimensional construct of the level of sexual desire, where sexual desire is defined as the interest in or desire to have sexual activity (Appendix A) [14,15]. The SDI-2 items were selected using the diagnostic criteria used in the DSM-III-R for Hypoactive Sexual Desire Disorder (HSDD) and clinical experience in the assessment and treatment of sexual desire disorders [14,15,16,17]. An SDI-2 score value ≤ 45 indicates reduced sexual desire.

Moreover, we used a second questionnaire (Figure 1), an unvalidated instrument created specifically for this study, to obtain the assessment of the influence of AD on the body image, sexuality, and self-perception of those affected. An 8-item questionnaire using a 0–10 scale (0–3 = no effect at all on patient’s SH, 3–6 = moderate effect on patient’s sexual health, 7–10 = severe effect on patient’s sexual health) was administered to all patients enrolled. Patients completed their questionnaire anonymously after the medical examination such that their answers could not be influenced by the physicians in any way. Dupilumab was administered at the baseline dose (600 mg as starting dose, followed by 300 mg every 2 weeks). This study was approved by the local ethical committee. 

## 3. Statistical Analysis

Quantitative variables were expressed as the mean and standard deviation (SD). Qualitative variables were expressed as frequencies and percentages. GraphPad Prism software (v 8.0; Graph Pad software, Inc., La Jolla, CA, USA) was used for all statistical analyses. The chi square test was used as appropriate to calculate statistical differences; a value of *p* < 0.05 was considered significant. Pearson’s correlation coefficient was used to evaluate the statistical relationship, or association, between two continuous variables.

## 4. Results

Three hundred and twenty-eight patients (170 males (51.8%); mean age: 33.78 ± 12.21 years (range 18–65)) met the inclusion criteria and were eligible for the study. The family statuses of patients (single, in a relationship, divorced, or separated) are reported in Table 1. AD had persisted since childhood (persistent AD) in 192/328 (58.5%) patients, while in 136/328 (41.5%), AD had directly started in adult age (≥18 years) (adult-onset AD). The average disease duration was 14.82 ± 10.33 years. The most frequently reported atopic comorbidity was rhinitis (91/328; 27.7%), followed by asthma (63/328; 19.2%), conjunctivitis (42/328; 12.8%), and food allergy (8/328; 2.4%), mostly anamnestic.

Baseline EASI was 26.84 ± 3.38 (median: 25), P-NRS was 8.89 ± 1.28 (median: 9), and DLQI was 22.8 ± 5.31 (median: 25). Flexural dermatitis was the most frequent clinical phenotype (121/328; 36.84%), followed by generalized eczema (93/328; 28.35%), prurigo nodularis-like (53/328; 16.15%), and hand eczema (42/328; 12.80%); other clinical phenotypes were less frequent (Table 1). The coexistence of more than one clinical phenotype was found in 31/328 (9.45%) patients. Involvement of the genital area and nipples was noted in 92/328 (28.04%; 50 males,42 females) and 31/328 patients (9.45%; 12 males,19 females), respectively. Regarding previous systemic treatment, CsA and systemic steroids were the most prescribed drugs (Table 1).

Dupilumab treatment led to a significant reduction in the mean EASI score from baseline to W4 and W16 (*p* < 0.0001 for all the assessments) (Figure 2 and Figure 3). Indeed, the average EASI percentage improvement with respect to the baseline was 57.56% and 92.13% at W4 and W16, respectively (Table 1). Overall, almost all patients (326/328; 99.3%) reached a 50% decrease in EASI (EASI-50), while EASI-75 and EASI-90 were reached by 78.04% (256/328) and 34.14% (112/328) of patients, respectively (Table 1).

Likewise, we observed a significant improvement in both mean P-NRS and DLQI from baseline at all timepoints (*p* < 0.0001) (Figure 2). The respective mean percentage reduction from baseline to W4 and W16 was 57.76% and 90.55% for P-NRS, and 60.52% and 76.75% for the DLQI score (Table 1).

The impact of AD on sexual desire assessed using SDI-2 showed a significant improvement during dupilumab treatment from the baseline to W4 and W16 (Figure 4). Indeed, SDI-2 had a mean value of 33.57 ± 8.18 (median: 41) at baseline vs. 46.34 ± 13.5 (median: 43) at W4 (*p* < 0.0001; mean percentage reduction 38.03%) and 56.34 ± 16.5 (median: 50) at W16 (*p* < 0.0001; mean percentage reduction 67.82%).

A higher burden of AD was found among patients with involvement of the genital area or nipples than those without, as assessed by DLQI (25.13 ± 2.28 vs. 20.8 ± 3.62; median 25 vs. 20; *p* < 0.0001) and SDI-2 (29.12 ± 5.64 vs. 37.42 ± 6.08; median 28 vs. 37; *p* < 0.0001).

Similar results were obtained with our questionnaire (Table 2). In particular, as regards the question “*Do you think your disease makes you less sexually attractive?*”, at baseline, 41.4% (*n* = 137) and 51.5% (*n* = 169) of patients reported that AD had a moderate (score: 4–6) (median 5) or severe effect (score: 7–10) (median: 9), respectively, compared to 8.5% (*n* = 28) (median: 4) and 1.5% (*n* = 5) (median: 7) at W16. Similarly, as regards the question “*Do you feel that your disease decreases your desire to seduce?*”, at baseline, 59.1% (*n* = 194) of subjects reported a moderate impact (median: 5) and 26.9% (*n* = 88) a severe impact (median: 9), respectively, with a statistically significant reduction at W16 of treatment (11.6% (*n* = 38; *p* < 0.0001) moderate impact (median: 4) and 2.4% (*n* = 8; *p* < 0.0001) severe impact (median: 7)). Interestingly, 46.9% (*n* = 154) of patients reported a severe impact (median: 9) of AD on the possibility of being seen naked by their partner at baseline, compared to 1.2% (*n* = 4; *p* < 0.0001) (median: 7) of patients at W16 (Table 2). Finally, as regards the question “*In your opinion, is your partner(s) afraid to have physical contact with you?*”, 42.7% (*n* = 140) and 48.5% (*n* = 159) reported a moderate (median: 6) and severe (median: 9) effect of the disease on their partner’s hesitation to engage in physical contact, respectively, with a statistically significant reduction at W16 (4.6% (*n* = 15; *p* < 0.0001) moderate (median: 4) effect and 0.6% (*n* = 2; *p* < 0.0001) severe (median: 7) impact).

There were no significant differences between the two sexes with regard to both tools used in our study.

## 5. Discussion

The assessment of SH impairment is usually poorly managed and evaluated in dermatological patients, mainly in patients with AD [4,9]. As a matter of fact, SH seems to be deeply affected by the disease; however, this aspect is underestimated, especially in women [16,17,18,19,20,21]. Many reasons could impair SH in these patients, e.g., the presence of lesions on the genital area, breasts or hands; psychiatric comorbidities; and allergic reactions to latex [9]. Furthermore, unpleasant sensations, such as itching, pain, and paresthesia, likely negatively influence SH beyond the lesions themselves; indeed, sensory disorders are very frequent in patients with AD [9].

Our data seem to confirm that moderate-to-severe AD significantly affects sexual desire, a particular aspect of SH, with a further negative impact on QoL. According to literature data, we found that the involvement of the genital and some erogenous areas such as the nipples was associated with a higher burden and more significant alterations in sexual desire [5,7,9].

Until now, and to the best of our knowledge, only one other study including 31 patients has assessed the positive impact of dupilumab therapy on sexual dysfunction [5]. The authors applied two different tools according to gender: the International Index Erectile Function (IIEF-5) in male patients and the Female Sexual Function Index for females [5]. After 6 months of therapy, an improvement in the sexual dysfunction index was observed, both in the male and female patients [5].

In our study, we evaluated the impact of both AD and dupilumab treatment on sexual desire and the influence of AD self-perception on mental attitudes towards sexual activity using SDI-2 and a second questionnaire especially created for the study. We found that the impact of AD on sexual desire showed a significant improvement during dupilumab treatment from the baseline to week 4 and week 16. This result was similar in both sexes, with no appreciable statistical difference.

The limitations of this study are that it was monocentric; it involved a relatively limited number of patients; the patients were followed for a short time; one of the questionnaires used was a not validated tool, and we did not include a control group.

Certain aspects, such as sexual attractiveness, sexual desire, the possibility of being seen naked by one’s partner, and a partner’s possible hesitation to engage in physical contact, have the greatest impact on patients’ lives and improve during therapy. Therefore, an early and effective treatment can significantly improve patients’ quality of life, including some aspects of SH.

## Figures and Tables

**Figure 1 medicina-58-01708-f001:**
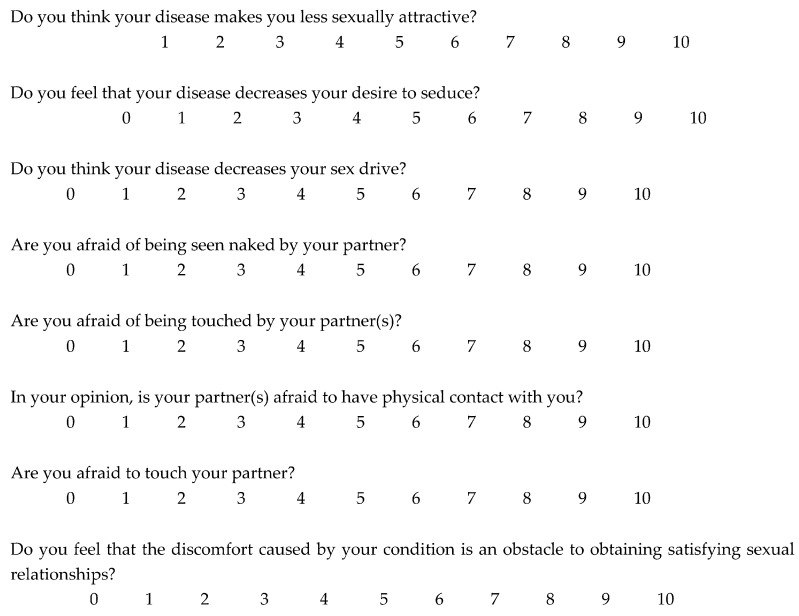
Unvalidated questionnaire on sexual health administered to 328 patients with moderate-to-severe atopic dermatitis.

**Figure 2 medicina-58-01708-f002:**
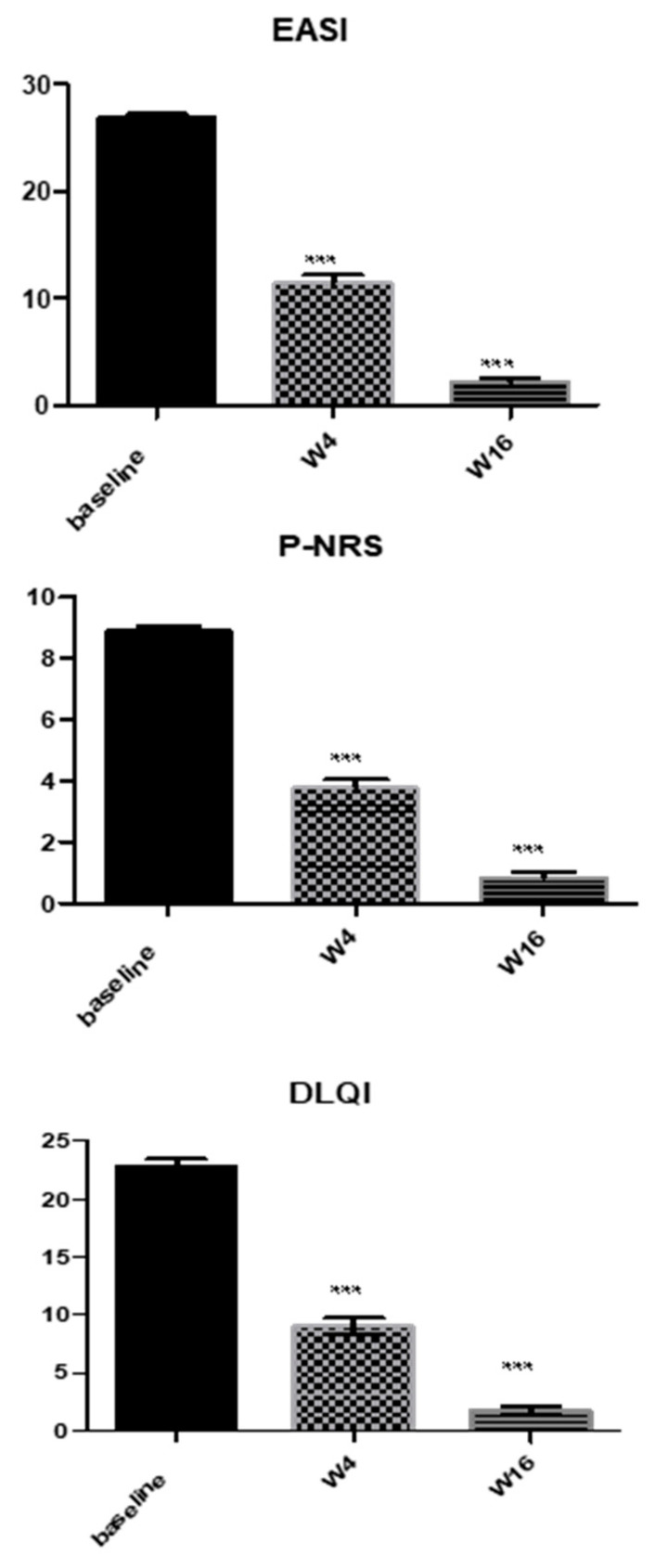
Mean variation in EASI, P-NRS, and DLQI from baseline to week 4 and week 16. Mean values of Eczema Area and Severity Index (EASI), Pruritus Numerical Rating Score (P-NRS), and Dermatology Life Quality Index (DLQI) of study population at baseline and after 4 weeks (W4) and W16 of dupilumab treatment. Statistical significance was assessed by the Mann–Whitney test and Fisher test: *** *p* < 0.0001.

**Figure 3 medicina-58-01708-f003:**
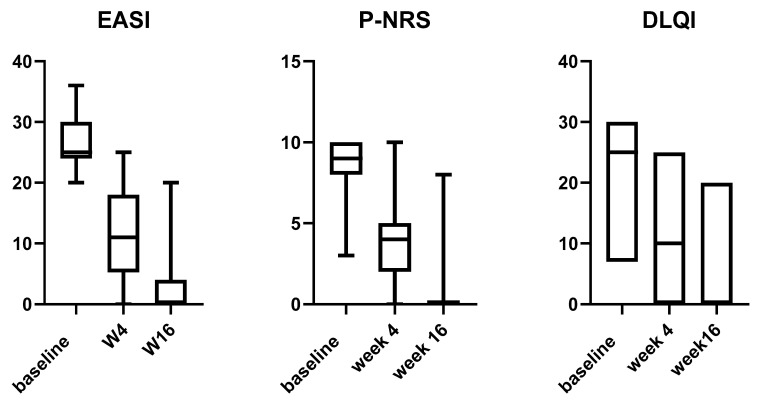
Box plot for EASI, P-NRS, and DLQI relative variations from baseline to week 16 in adult patients treated with dupilumab. EASI: Eczema Area and Severity Index; P-NRS: Pruritus Numerical Rating Score; DLQI: Dermatology Life Quality Index.

**Figure 4 medicina-58-01708-f004:**
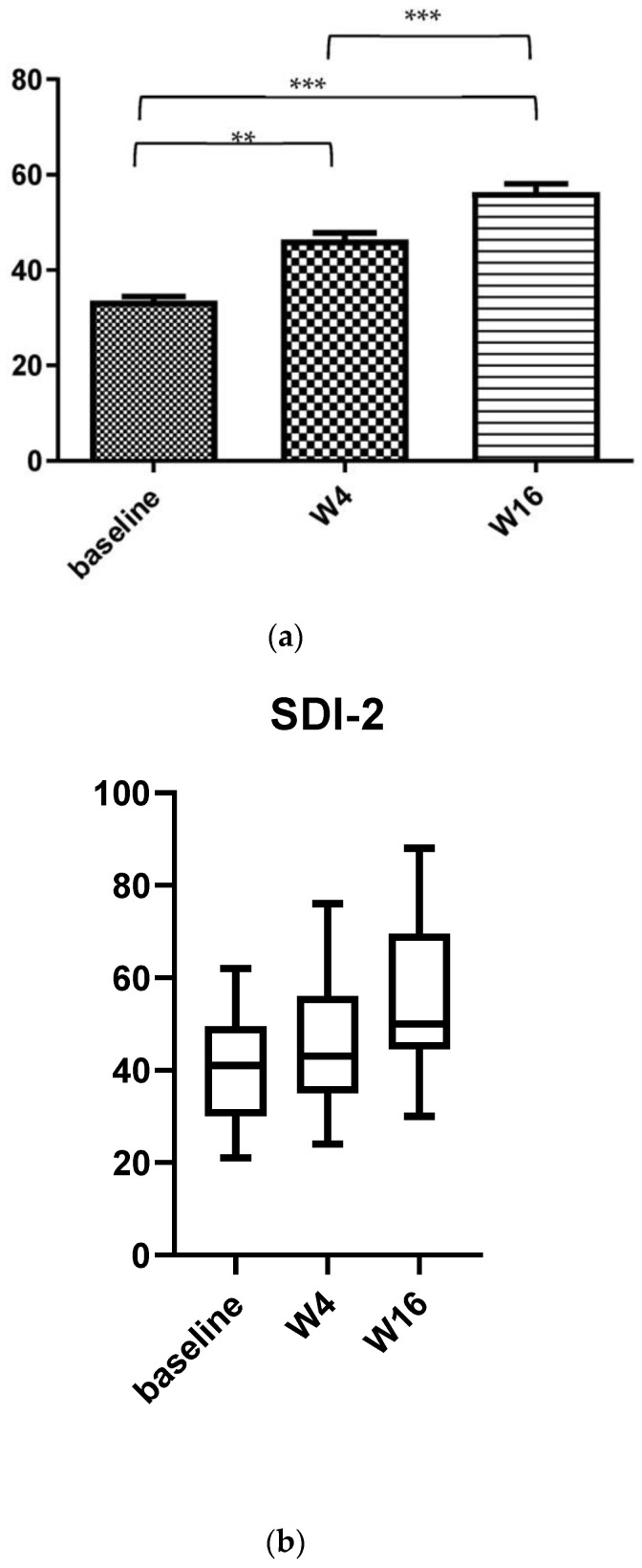
(**a**) Mean variation in SDI-2 from baseline to week 4 and week 16 during dupilumab treatment. (**b**) Box plot for SDI-2 relative variations from baseline to week 4 and 16 in adult patients treated with dupilumab. (**b**) Mean values of Sexual Desire Inventory-2 (SDI-2) of study population at baseline and after 4 weeks (W4) and W16 of dupilumab treatment. Statistical significance was assessed by the Mann–Whitney test and Fisher test: *** *p* < 0.0001, ** *p* < 0.001,.

**Table 1 medicina-58-01708-t001:** Demographic and clinical characteristics of patients.

Number of Patients	328
**Sex**	
Male	170 (51.8%)
Female	158 (48.2%)
**Family status**	
Single	102 (31.1%)
In a relationship	138 (42.1%)
Divorced or separated	88 (26.8%)
**Occurrence**	
Persistent	192 (58.5%)
Adult-onset	136 (41.5%)
**Average disease duration (years)**	14.82 ± 10.33
**Atopic comorbidities**	
Rhinitis	91 (27.7%)
Asthma	63 (19.2%)
Conjunctivitis	42 (12.8%)
Food allergy	8 (2.4%)
**Clinical phenotype**	
Flexural dermatitis	121 (36.8%)
Generalized eczema	93 (28.4%)
Prurigo nodularis-like	53 (16.2%)
Hand eczema	42 (12.8%)
Nummular eczema	6 (1.8%)
Head and neck eczema	3 (0.9%)
**Involvement of genital area**	92 (28.04%)
**Involvement of nipples**	31 (9.45%)
**Previous systemic treatment**	
Cyclosporine	125 (38.1%)
Systemic steroids	98 (29.9%)
Others	13 (4%)
**AD severity score *(mean ± SD; median)***	
**Baseline**	
EASI	26.84 ± 3.38; 25
P-NRS	8.89 ± 1.28; 9
DLQI	22.8 ± 5.31; 25
**Week 4**	
EASI	11.39 ± 6.83; 11 (*p* < 0.0001)
P-NRS	3.76 ± 2.49; 4 (*p* < 0.0001)
DLQI	9 ± 6.49; 10 (*p* < 0.0001)
**Week 16**	
EASI	2.11 ± 3.82; 0 (*p* < 0.0001)
P-NRS	0.84 ± 1.75; 0 (*p* < 0.0001)
DLQI	1.75 ± 3.54; 0 (*p* < 0.0001)
**Average EASI percentage improvement**	
After 4 weeks	57.56%
After 16 weeks	92.13%
**Average P-NRS percentage improvement**	
After 4 weeks	57.76%
After 16 weeks	90.55%
**Average DLQI percentage improvement**	
After 4 weeks	60.52%
After 16 weeks	76.75%
**SDI-2 (mean value ± SD; median)**	
At baseline	33.57 ± 8.18; 41
After 4 weeks	46.34 ± 13.5; 43
**After 16 weeks**	56.34 ± 16.5; 50

EASI: Eczema Area Severity Index; P-NRS: Pruritus Numeric Rating Scale; DLQI: Dermatology Life Quality Index; SDI-2: Sexual Desire Inventory-2.

**Table 2 medicina-58-01708-t002:** Results of an unvalidated questionnaire on sexual health in 328 adult patients with moderate-to-severe atopic dermatitis.

	*Baseline* *Median; % of Patients (Number of Patients)*			*Week 4 (W4)* *Median; % of Patients (Number of Patients)*			*Week 16 (W16)* *Median; % of Patients (Number of Patients)*		
	Score 0–3	Score: 4–6	Score 7–10	Score 0–3	Score 4–6	Score 7–10	Score 0–3	Score 4–6	Score 7–10
Q1	2; 6.7% (*n* = 22)	5; 41.8% (*n* = 137)	9; 51.5% (*n* = 169)	1; 50.6% *** (*n* = 166)	5; 43.3% ^NS^ (*n* = 142)	8; 6.1% *** (*n* = 20)	1; 89.9% *** (*n* = 295)	4; 8.5% *** (*n* = 28)	7; 1.5% *** (*n* = 5)
Q2	2; 14% (*n* = 46)	5; 59.1% (*n* = 194)	9; 26.9% (*n* = 88)	0; 53.7% *** (*n* = 176)	4; 44.2% * (*n* = 145)	7; 2.1% *** (*n* = 7)	1; 86% *** (*n* = 282)	4; 11.6% *** (*n* = 38)	7; 2.4% *** (*n* = 8)
Q3	2; 48.2% (*n* = 158)	5; 47.6% (*n* = 156)	9; 4.2% (*n* = 14)	2; 55.5% ^NS^ (*n* = 182)	5; 43.6% ^NS^ (*n* = 143)	8; 0.9% ** (*n* = 3)	1; 91.5% *** (*n* = 300)	5; 7.9% *** (*n* = 26)	7; 0.6% ** (*n* = 2)
Q4	2; 9.7% (*n* = 32)	5; 43.3% (*n* = 142)	9; 46.9% (*n* = 154)	2; 16.2% ^NS^ (*n* = 53)	5; 57.9% ^NS^ (*n* = 190)	9; 25.9% *** (*n* = 85)	1; 91.2% *** (*n* = 299)	4; 7.6% *** (*n* = 25)	7; 1.2% *** (*n* = 4)
Q5	3; 14.3% (*n* = 47)	5; 59.5% (*n* = 195)	9; 26.2% (*n* = 86)	1; 51.8% *** (*n* = 170)	5; 43.6% ^NS^ (*n* = 143)	9; 4.6% *** (*n* = 15)	1; 93.6% *** (*n* = 307)	6; 5.8% *** (*n* = 19)	7; 0.6% *** (*n* = 2)
Q6	3; 8.8% (*n* = 29)	6; 42.7% (*n* = 140)	9; 48.5% (*n* = 159)	1; 22.2% *** (*n* = 73)	4; 62.5% ** (*n* = 205)	8; 27.4% *** (*n* = 90)	1; 94.8% *** (*n* = 311)	4; 4.6% *** (*n* = 15)	7; 0.6% *** (*n* = 2)
Q7	3; 51.2% (*n* = 168)	6; 42.7% (*n* = 140)	9; 6.1% (*n* = 20)	1; 94.5% *** (*n* = 310)	4; 4.9% *** (*n* = 16)	8; 0.6% *** (*n* = 2)	1; 96.3% *** (*n* = 316)	4; 3.4% *** (*n* = 11)	7; 0.3% *** (*n* = 1)
Q8	2; 7.6% (*n* = 25)	6; 41.8% (*n* = 137)	9; 50.6% (*n* = 166)	0; 26.5% *** (*n* = 87)	4; 59.1% * (*n* = 194)	7; 14.3% *** (*n* = 47)	1; 93.3% *** (*n* = 306)	4; 5.8% *** (*n* = 19)	7; 0.9% * (*n* = 3)

**Patient questionnaire:** Q1: Do you think your disease makes you less sexually attractive?; Q2: Do you feel that your disease decreases your desire to seduce?; Q3: Do you think your disease decreases your sex drive?; Q4: Are you afraid of being seen naked by your partner?; Q5: Are you afraid of being touched by your partner(s)?; Q6: In your opinion, is your partner(s) afraid to have physical contact with you?; Q7: Are you afraid to touch your partner?; Q8: Do you feel that the discomfort caused by your condition is an obstacle to obtaining satisfying sexual relationships? Score 0–3 = no effect; score 3–6 = moderate effect; score 7–10 = severe effect. Statistical significance was assessed by chi square test: *** *p* < 0.0001, ** *p* < 0.001, * *p* < 0.01, NS: not significant.

## Data Availability

Data sharing not applicable to this article as no datasets were generated or analyzed during the current study.

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
