# Peer review of "Effect of Dupilumab on Sexual Desire in Adult Patients with Moderate to Severe Atopic Dermatitis"

_medicina, 2022, doi:10.3390/medicina58121708_

Round 1
Reviewer 1 Report
Sexual function and the impact on atopic dermatitis are often neglected clinically.This article points out the views of the impact of AD on sexual desire, and the association of improvement after treatment.
In the discussion , the author mentioned that there is an improvement in the sexual dysfunction index , both in the male and female patients .Can the author provide any informations of the difference in details between male and female patients ?
Author Response
Dear Editor and Reviewers
We thank for the interest in our paper.
We have read your comments with great interest since they improve our work and have made the changes as requested.
Below are our point-by-point responses:
REVIEWER 1
No significant differences were found between the two sexes with regard to the variables considered. This information was already reported in the ‘Results’ paragraph of the submitted paper. However, as recommended by the reviewer, the concept has been better defined and also included in the Abstract and Discussion paragraphs.
Reviewer 2 Report
The article titled ‘Effect of Dupilumab on Sexual Desire in adult patients with moderate to severe atopic dermatitis’ may be an useful contribution to the journal; however, few changes should be taken into consideration:
Please elaborate on including patients unde 18 years old in respect to legal age consent and all ethical issues related to it in the country where the study was realised
Limitations of the study should be inserted as a separate paragraph (e.g. unicentricity of the study, etc).
Graphical box and whisker plots could be included, as it would be in the interest if the reader at least for most important scales (SDI-2, for example), to offer a clearer and readier image of the findings of the study.
Also, wherever the data is not normally distributed, the authors should introduce median, not the mean, as in currrent form (e.g. in Table 1).
Statistical analysis should be elaborated on, in respect to use of parametric and non-parametric tests.
Grammar and punctuation must also be carefully checked within the entire article.
Author Response
Dear Editor and Reviewers
We thank for the interest in our paper.
We have read your comments with great interest since they improve our work and have made the changes as requested.
Below are our point-by-point responses:
REVIEWER 2
As already reported in ‘Material and Methods’ paragraph, only patients 18-year-old or older were enrolled. They all signed an informed consent approved by the local Ethical Committee. This information has been added to the study.
Limitations of the study have been added in ‘Discussion’ paragraph.
Graphical box and whisker plots for SDI-2 and other relevant variables results have been added to the paper.
Median values have been reported both in the tables and the text.
Statistical analysis has been reworked for all the data, the text and the data have been modified accordingly.
The English language has been reviewed by a native-speaker consultant.